# Identification of circadian clock modulators from existing drugs

T Katherine Tamai[1], Yusuke Nakane[1,2], Wataru Ota[1,2], Akane Kobayashi[1,2], Masateru Ishiguro[1,2], Naoya Kadofusa[1], Keisuke Ikegami[3], Kazuhiro Yagita[4], Yasufumi Shigeyoshi[3], Masaki Sudo[1], Taeko Nishiwaki-Ohkawa[1,2], Ayato Sato[1] & Takashi Yoshimura[1,2,5,6,*]

## Abstract

Chronic circadian disruption due to shift work or frequent travel across time zones leads to jet-lag and an increased risk of diabetes, cardiovascular disease, and cancer. The development of new pharmaceuticals to treat circadian disorders, however, is costly and hugely time-consuming. We therefore performed a high-throughput chemical screen of existing drugs for circadian clock modulators in human U2OS cells, with the aim of repurposing known bioactive compounds. Approximately 5% of the drugs screened altered circadian period, including the period-shortening compound dehydroepiandrosterone (DHEA; also known as prasterone). DHEA is one of the most abundant circulating steroid hormones in humans and is available as a dietary supplement in the USA. Dietary administration of DHEA to mice shortened free-running circadian period and accelerated re-entrainment to advanced light–dark (LD) cycles, thereby reducing jet-lag. Our drug screen also revealed the involvement of tyrosine kinases, ABL1 and ABL2, and the BCR serine/threonine kinase in regulating circadian period. Thus, drug repurposing is a useful approach to identify new circadian clock modulators and potential therapies for circadian disorders.

**Keywords** circadian rhythms; DHEA; drug repurposing; jet-lag; tyrosine kinases

**Subject Categories** Metabolism; Pharmacology & Drug Discovery

## Introduction

Circadian clocks are highly conserved, endogenous timers present in virtually all living organisms. These clocks regulate near 24-h rhythms in numerous behavioral and physiological processes, including our sleep–wake cycles and metabolism. In mammals, these daily rhythms are controlled by a central circadian pacemaker, the suprachiasmatic nucleus (SCN), located in the anterior hypothalamus (Ralph *et al*, 1990; Klein *et al*, 1991). Most peripheral tissues and cells also contain self-sustained circadian oscillators (Balsalobre *et al*, 1998; Yamazaki *et al*, 2000; Yoo *et al*, 2004), which are driven by transcriptional–translational feedback loops composed of circadian clock genes and proteins (Takahashi, 2015). The basic helix-loop-helix proteins CLOCK and BMAL1 heterodimerize to form a transcriptional activator complex and activate the *PER* and *CRY* repressor genes, whose protein products, in turn, repress their own transcription.

Disruption of the circadian clock due to shift work or travel across time zones leads to circadian desynchrony, or jet-lag, and reflects a mismatch between the internal biological clock and external time cues (Arendt, 2009). Chronic circadian misalignment has long-term consequences on our health and often leads to an increased risk of diabetes, cardiovascular disease and cancer (Davidson *et al*, 2006; Scheer *et al*, 2009; Buxton *et al*, 2012). Recent circadian studies have identified several genes that by mutation, chemical inhibition or knockdown, significantly reduce jet-lag, including the vasopressin V1a and V1b receptors, casein kinase 1 epsilon (CK1ε), and salt inducible kinase 1 (Sik1; Jagannath *et al*, 2013; Yamaguchi *et al*, 2013; Pilorz *et al*, 2014). Injection of the neuropeptide vasoactive intestinal polypeptide (VIP) into the mouse SCN also accelerates re-entrainment to new light–dark (LD) cycles (An *et al*, 2013). However, there are currently no orally available drugs or supplements, except melatonin perhaps, to combat the complex behavioral and metabolic consequences of jet-lag.

The development of new pharmaceuticals, from drug discovery to market approval, is costly and hugely time-consuming with a high rate of failure. Drug repurposing, or identifying new functions for existing compounds, is therefore a popular approach to fast track potential drugs through to clinical trials (Rennekamp & Peterson, 2015). We used a high-throughput chemical screening strategy to

1 Institute of Transformative Bio-Molecules (WPI-ITbM), Nagoya University, Nagoya, Japan
2 Laboratory of Animal Physiology, Graduate School of Bioagricultural Sciences, Nagoya University, Nagoya, Japan
3 Department of Anatomy and Neurobiology, Kindai University Faculty of Medicine, Osaka, Japan
4 Department of Physiology and Systems Bioscience, Kyoto Prefectural University of Medicine, Kyoto, Japan
5 Avian Bioscience Research Center, Graduate School of Bioagricultural Sciences, Nagoya University, Nagoya, Japan
6 Division of Seasonal Biology, National Institute for Basic Biology, Okazaki, Japan
*Corresponding author. Tel: +81 52 789 4056; E-mail: takashiy@agr.nagoya-u.ac.jp

isolate new circadian clock modulators (Hirota *et al*, 2008; Isojima *et al*, 2009; Chen *et al*, 2011), with the aim of repurposing known bioactive compounds. We tested over 1,000 small molecules from two chemical libraries, including US Food and Drug Administration (FDA)-approved drugs and the International Drug Collection (IDC), and we identified nearly 60 chemicals that altered circadian period, including the period-shortening molecule DHEA. When tested in mice *in vivo*, DHEA significantly shortened circadian period of locomotor rhythms and accelerated re-entrainment to advanced LD cycles, suggesting that DHEA might serve as a convenient over-the-counter treatment for jet-lag. Our chemical screen also identified several tyrosine kinase inhibitors that modulated clock function, implicating this class of kinases in the control of circadian period. Thus, drug repurposing is a useful strategy to identify potential treatments for circadian misalignment and new clock regulators.

# Results

## Identification of new circadian clock modulators

Using a human osteosarcoma U2OS cell line stably expressing the clock reporter *Bmal1-dLuc* (Oshima *et al*, 2015), we screened over 1,000 molecules from an FDA-approved drug library and the IDC to identify new circadian clock modulators. To reduce the number of false positives, all compounds were tested in triplicate at two different concentrations (10 and 1 μM) in a 384-well plate format. Primary screening identified several potential hit compounds based on period change. Drugs that lengthened circadian period by 1 or more hours or shortened period by 0.5 or more hours (Fig 1A) were selected for additional screening, although some chemicals, whose effects were clearly dose-dependent but below threshold values, were also included. Of these 72 potential hit compounds, 59 were validated in a secondary screen for dose-dependency (Appendix Fig S1), with 46 chemicals lengthening and 13 compounds shortening circadian period. Thus, ~5% of drugs currently on the market or in clinical trials altered circadian period. This indicates that, depending on the dose and duration of treatment for other ailments, one "side effect" of these drugs might be perturbation of the circadian clock. Therapeutic classification revealed eight different categories of hit compounds, including (i) anti-cancer and immunosuppressive drugs, (ii) disinfectants and antiseptics, (iii) hormones and contraceptives, (iv) drugs targeting the central nervous system, (v) dermatological agents, (vi) gastrointestinal drugs, (vii) vitamins and minerals, and (viii) cardiovascular agents (Fig 1B). Importantly, some of these compounds have been previously identified in circadian drug screens (i.e., DNA damage agent, mitoxantrone; Hirota *et al*, 2008), whereas others affect pathways that have been implicated in regulating circadian rhythms (i.e., EGF receptor inhibitor, erlotinib; Kramer *et al*, 2001), thus validating our screening protocol. One obvious advantage of screening existing drugs is that they frequently have known mechanisms of action (Rennekamp & Peterson, 2015). All known targets of hit compounds identified in our screen are listed in Table EV1. In addition, the top six categories of drug targets are shown and include DNA, the androgen receptor (AR), DNA topoisomerase 2α, the nuclear receptor subfamily 1 group I member 2 (NR1I2), the retinoic acid receptor α (RARα)/ retinoid X receptor β (RXRβ), and the tubulin β chain (Fig 1C).

## DHEA primarily shortens circadian period

Relatively few chemicals identified in our screen shortened circadian period, so we initially focused on characterizing some of these compounds. Interestingly, the steroid hormone DHEA and its derivative DHEA acetate showed clear dose-dependent shortening of circadian period in U2OS cells (Fig 2A and Appendix Fig S1). Recent reports, however, have shown that high doses of DHEA (e.g., 100 μM) lengthen circadian period and decrease amplitude (Rey *et al*, 2016; Putker *et al*, 2017), which we have confirmed (Fig EV1). In humans, DHEA is a steroid hormone produced in the adrenal gland, gonads, and brain and is one of the most abundant circulating hormones. It functions as an androgen precursor and can be converted to testosterone or estrogen (Appendix Fig S2). To determine whether DHEA itself or one of its precursors or metabolites modulates circadian period, we treated U2OS cells with several endogenous steroid hormones and monitored bioluminescence. These results show that deoxycorticosterone (21-hydroxyprogesterone), 11-deoxycortisol, and testosterone lengthened circadian period (Appendix Fig S2). Some of these hormones were included in our primary drug screen, but were not identified as hit compounds, because their effects on circadian period were below threshold values. In contrast, progesterone and 17α-hydroxyprogesterone lengthened period by more than 1 h and were indeed classified as hit compounds (Appendix Figs S1 and S2). Interestingly, DHEA was the only endogenous steroid hormone tested that shortened period in U2OS cells. Notably, its metabolite DHEA sulfate had no effect on circadian period (Appendix Fig S2).

To determine whether the effects of DHEA on the circadian clock were cell type- or tissue-specific, we treated $mPer2^{Luc}$ mouse embryonic fibroblasts (MEFs) with DHEA. Similar to our observations in U2OS cells, DHEA shortened circadian period in MEFs (Fig 2B). We then prepared explant cultures of SCN and lung from $mPer2^{Luc}$ mice and treated them with DHEA. Although slightly higher concentrations were required, we observed significant shortening of circadian period in both of these tissues (Fig 2B). These results indicated that DHEA indeed shortens circadian period in cells and tissues, including the SCN.

Numerous studies in mice have shown that DHEA can be administered orally (Milewich *et al*, 1995). Therefore, to test its effects *in vivo*, we added DHEA directly to powdered food and assessed circadian clock function by monitoring wheel-running activity. We also surgically implanted temperature devices into mice to measure body temperature rhythms. Mice were initially fed powdered food without drug and exposed to a light–dark cycle (12L:12D) for 1 or 2 weeks to entrain their circadian clocks (Fig 2C). Animals were then transferred into constant darkness (DD) for about 1 week to measure free-running circadian period. Mice were then given either normal powdered food or powdered food mixed with 0.5% DHEA (w/w) for 6 days. This dose was then increased to 1% DHEA (w/w) for another 6 days. Importantly, these concentrations of DHEA have been previously tested in mice and shown to be safe (Milewich *et al*, 1995). Mice were then returned to normal powdered food until the end of the experiment. These results show that DHEA had a profound effect on behavior and appeared to shorten circadian period of both activity and body temperature rhythms in a dose-dependent manner (Figs 2C and EV2, respectively). Surprisingly, after return to

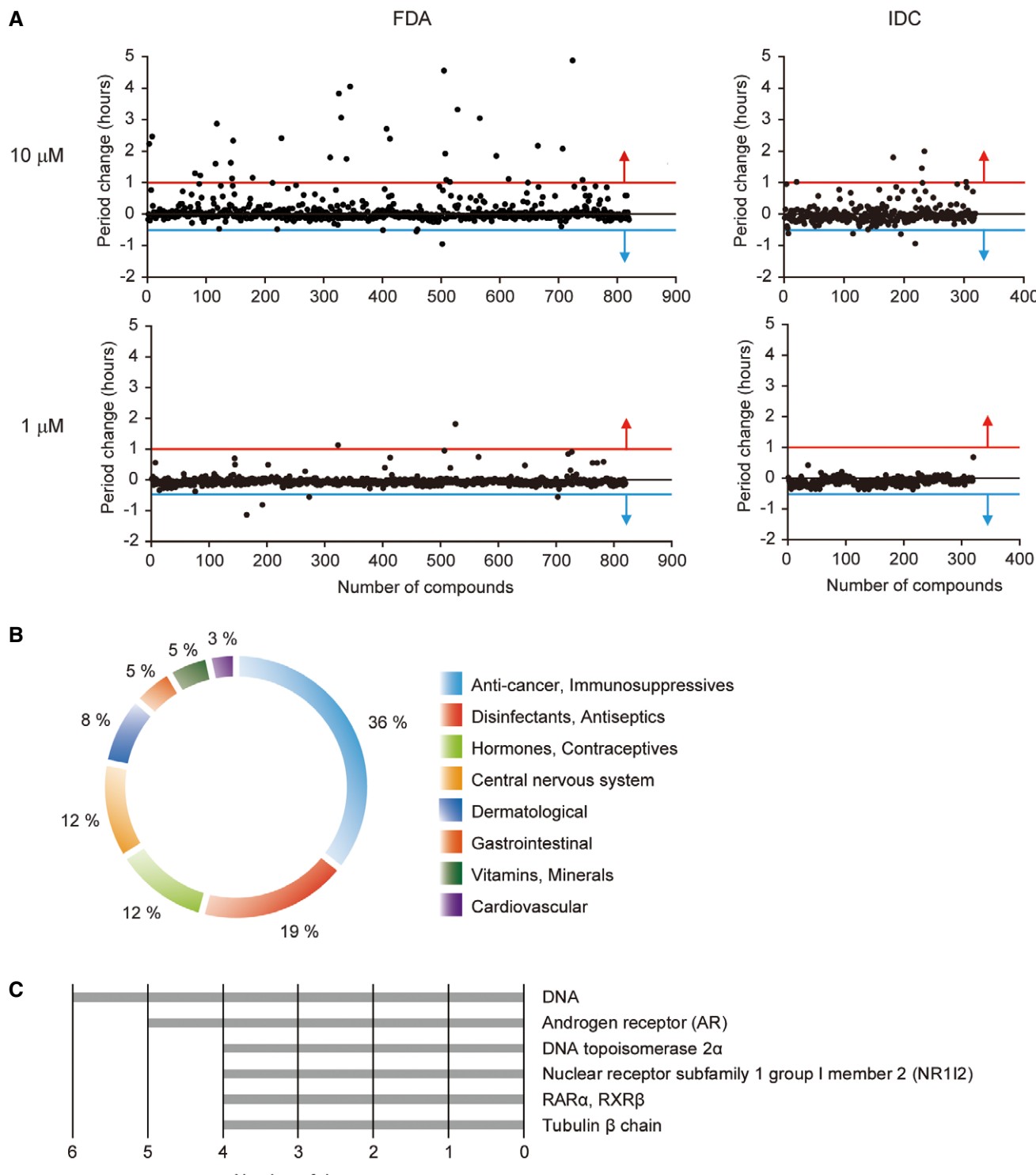

**Figure 1. Chemical screening for circadian clock modulators.**

A   US Food and Drug Administration (FDA)-approved drugs (left graphs) and the International Drug Collection (IDC) (right graphs) were screened in U2OS clock reporter cells at 10 μM (top graphs) and 1 μM (bottom graphs). Primary screening identified 72 potential hit compounds that lengthened (by 1 or more hours; at or above red line) or shortened (by 0.5 or more hours; at or below blue line) circadian period.

B   Therapeutic classification of 59 hit compounds validated in a secondary screen for dose-dependency (data for all 59 hit compounds are shown in Appendix Fig S1, and all statistical information is shown in Appendix Table S2).

C   Top six targets of hit compounds reported in the DrugBank or KEGG DRUG databases.

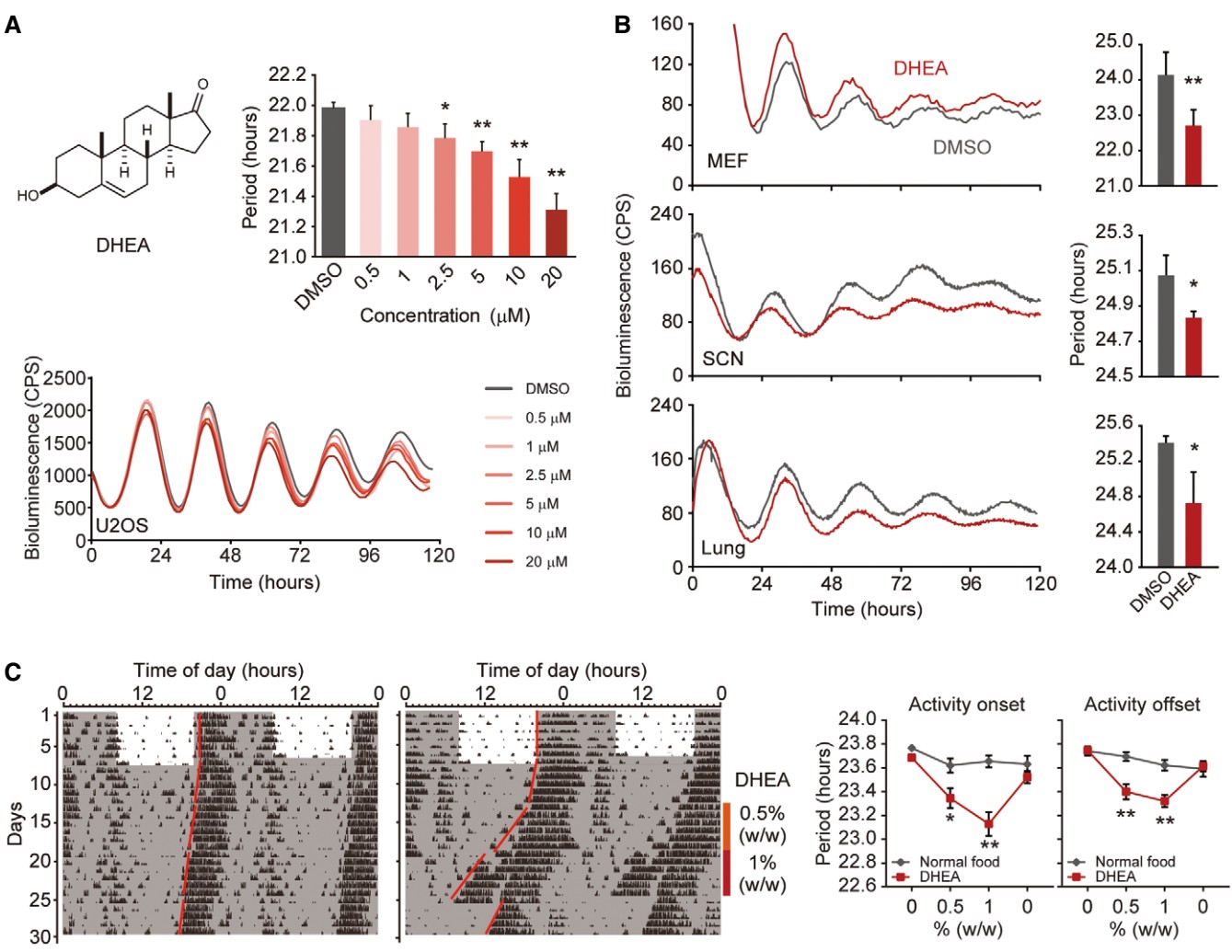

**Figure 2. DHEA shortens circadian period.**

A Chemical structure of DHEA (left) and its dose-dependent effects on circadian period in U2OS cells (histogram, right; luminescent traces, below). Data are the mean ± SEM from three independent experiments and were analyzed by one-way ANOVA, followed by a Dunnett's test (*$P < 0.05$, **$P < 0.01$). All statistical information is shown in Appendix Table S1.

B Effect of 20 μM DHEA on mouse embryonic fibroblasts (MEFs) and 50 μM DHEA on SCN and lung from *mPer2^Luc* mice. Data are presented as the mean ± SEM of four or five independent experiments and were analyzed by a Welch's *t*-test (*$P < 0.05$, **$P < 0.01$). All statistical information is shown in Appendix Table S1.

C Representative double-plotted actograms of control ($n = 8$; left) and DHEA-treated ($n = 14$; middle) animals entrained in LD and transferred into DD. DHEA (0.5% w/w; vertical line) was administered in powdered food ~1 week after transfer into DD for 6 days and then increased to 1.0% (w/w) for another 6 days. Animals were then returned to normal powdered food without drug for 1 week. Free-running period was calculated based on activity onset (left graph) or activity offset (right graph) and plotted as the mean ± SEM (far right). Data were analyzed by two-way ANOVA, followed by a Sidak's multiple comparisons test (*$P < 0.05$, **$P < 0.01$). All statistical information is shown in Appendix Table S1.

normal powdered food, activity onset returned to its original phase (i.e., to the phase before drug treatment; Fig 2C). Although DHEA is known to readily cross the blood–brain barrier (Stárka *et al*, 2015), these data suggest that its effects on behavior were independent of the SCN clock and might be acting through extra-SCN circadian pacemaker(s) (Pezuk *et al*, 2010).

### Dietary administration of DHEA reduces jet-lag in mice

By temporarily speeding up or slowing down the circadian clock, drugs that shorten or lengthen period, respectively, might accelerate

re-entrainment to new light–dark (LD) cycles, thereby reducing jet-lag (Fig EV3). To test this hypothesis, we performed a jet-lag experiment and examined the effects of DHEA on re-entrainment to a 6-h phase-advanced LD cycle (Fig 3). We fed mice normal powdered food, as above, and exposed them to a LD cycle to synchronize their circadian clocks. After ~1 week, mice were given either normal powdered food or food mixed with DHEA, and the LD cycle was advanced by 6 h (Fig 3A). Chronic administration of DHEA led to a dramatic phase advance in activity onset compared to controls. In drug-treated mice, this advance occurred almost immediately to the new LD cycle, but surprisingly, subsequently appeared to free-run,

even under LD conditions (Fig 3A). In some mice, this phase advance in activity onset appeared to stabilize ~4–6 h prior to lights off (Fig EV4). We wondered whether adjusting the treatment time of DHEA might be sufficient to reduce jet-lag to a 6-h phase-advanced LD cycle. Therefore, in the next experiment, treatment with DHEA was limited to the first 3 days of the new LD cycle (Fig 3B). Mice were then returned to normal food for the remainder of the experiment. These results show that mice given an acute dose of DHEA shifted much more rapidly to the new LD cycle and re-entrained within 1–3 days compared to the 6–7 days required for control mice (Fig 3B). In addition, body temperature rhythms in DHEA-treated animals appeared to shift and reset more rapidly to the new LD cycle compared to controls (the trough more so than the peak; Fig 3C). To determine whether acute DHEA treatment did, in fact, advance the circadian clock, we again treated mice with DHEA and exposed them to an advanced LD cycle for 3 days. Animals were then returned to normal food and transferred into DD (Fig 3D). These results show that the clock has indeed shifted more in DHEA-treated animals than in control mice. Thus, by adjusting the treatment time of DHEA to 3 days, it was possible to phase shift the circadian clock and accelerate re-entrainment to an advanced LD cycle, thereby reducing jet-lag with dietary DHEA.

### Involvement of ABL and BCR kinases in the circadian clockwork

Two tyrosine kinase inhibitors were identified in our screen, dasatinib and nilotinib, that lengthened and shortened circadian period, respectively, and both were effective in the nanomolar range (Fig 4A). These drugs are second-generation inhibitors of the BCR-ABL tyrosine kinase and approved for treatment of chronic myelogenous leukemia (CML; Molica et al, 2017). Most cases of CML are caused by a chromosomal translocation, called the Philadelphia chromosome, in which the break point cluster (BCR) gene on chromosome 22 and the Abelson (ABL) non-receptor type tyrosine kinase gene on chromosome 9 are fused, resulting in the chimeric oncogene, BCR-ABL (Faderl et al, 1999). Although much is known about the consequences of the BCR-ABL fusion protein in CML, the function of BCR and ABL in intact animals is not well understood (Schwartzberg et al, 1989; Tybulewicz et al, 1991; Voncken et al, 1996).

Nilotinib is a selective BCR-ABL tyrosine kinase inhibitor, while dasatinib is a multi-targeted inhibitor of BCR-ABL tyrosine kinase and SRC family kinases (Ciarcia et al, 2016; Marfe et al, 2016). There is also some evidence that dasatinib may inhibit the circadian clock kinase CK1 (Karaman et al, 2008), which may explain its period-lengthening effects (Fig 4A). When we examined additional BCR-ABL tyrosine kinase inhibitors, we found that the more selective inhibitors, imatinib and bafetinib, acted similarly to nilotinib and shortened circadian period, while the less selective, multi-targeted inhibitors bosutinib and ponatinib, lengthened circadian period (Fig EV5).

To determine whether the ABL1, ABL2, BCR, and SRC kinases are indeed the targets of these chemical inhibitors, we next examined the effect of siRNA-mediated knockdown of each of these genes in U2OS cells. Knockdown of either ABL1 or BCR shortened circadian period (Fig 4B and Appendix Fig S3), consistent with previous results from a large-scale circadian RNAi screen (Zhang et al, 2009). siRNAs to ABL2 also shortened circadian period, but those to SRC

had no significant effect (Fig 4B and Appendix Fig S3). Together, these results indicate that ABL1, ABL2, and BCR are possible targets for the period-shortening inhibitors, nilotinib, imatinib, and bafetinib, and implicate these kinases in the regulation of circadian period (Fig 4B). Interestingly, the Abl1, Abl2, and Bcr genes, and even Src, are expressed in the mouse SCN, although their expression does not appear to be rhythmic (Fig 4C and Appendix Fig S4).

## Discussion

Due to the high cost and time-consuming nature of developing new pharmaceuticals, drug repurposing approaches have become increasingly popular. With this strategy in mind, we screened over 1,000 existing drugs for new circadian clock modulators. This led to the discovery of 59 period-changing compounds, including the steroid hormone DHEA, which primarily shortened circadian period and accelerated re-entrainment to advanced LD cycles in mice. DHEA is one of the most abundant circulating steroid hormones in humans; however, its circulating levels are significantly lower in mice than in humans (0.28 versus 1.83 nM, respectively). Therefore, it is not yet known whether the results from our study could be translated to humans, particularly the high, albeit safe, doses used in mice. Clearly, further investigation is required to evaluate the effectiveness of DHEA for the treatment of jet-lag in humans.

In humans, DHEA is produced by the adrenal gland, gonads, and brain. Studies in primates have shown that, like cortisol, adrenal secretion of DHEA displays a robust circadian rhythm and peaks in the early day (Lemos et al, 2006). This steroid hormone appears to function primarily as a precursor to testosterone and estrogen. Indeed, previous circadian studies in mice have revealed that treatment with testosterone leads to significant changes in locomotor behavior, including shortening of circadian period and increased wheel-running activity (Daan et al, 1975). This indicates that some of the effects we observed with DHEA in vivo might be due, in part, to its conversion to testosterone. Surprisingly, however, testosterone treatment of U2OS cells lengthened circadian period, reflecting clear in vivo versus in vitro differences, which we do not yet fully understand. Interestingly, of the endogenous steroids tested, DHEA was the only one that shortened circadian period in U2OS cells.

Dehydroepiandrosterone has been the subject of numerous investigations due its protective effects from a variety of disorders, including obesity and diabetes to cardiovascular disease and cancer (Milewich et al, 1995). More recent studies have focused on its role in aging, and it is currently being sold in the USA as an anti-aging supplement (Baulieu et al, 2000; Corona et al, 2013). Reports have shown that DHEA levels in humans steadily decline with age and that treatment of elderly subjects with 50 mg/day over 1 year improved general health and well-being without harmful consequences (Baulieu et al, 2000). Although DHEA appears to be relatively safe, whether it could be used to treat circadian disorders in humans, as mentioned above, requires further investigation.

Previous circadian studies have identified several genes, chemicals, and signaling pathways involved in reducing jet-lag. One important factor in accelerating re-entrainment to new LD cycles appears to be the reduction of intercellular communication between SCN neurons. Yamaguchi et al (2013) reported that mice lacking the vasopressin receptors V1a and V1b ($V1a^{-/-} V1b^{-/-}$) were resistant

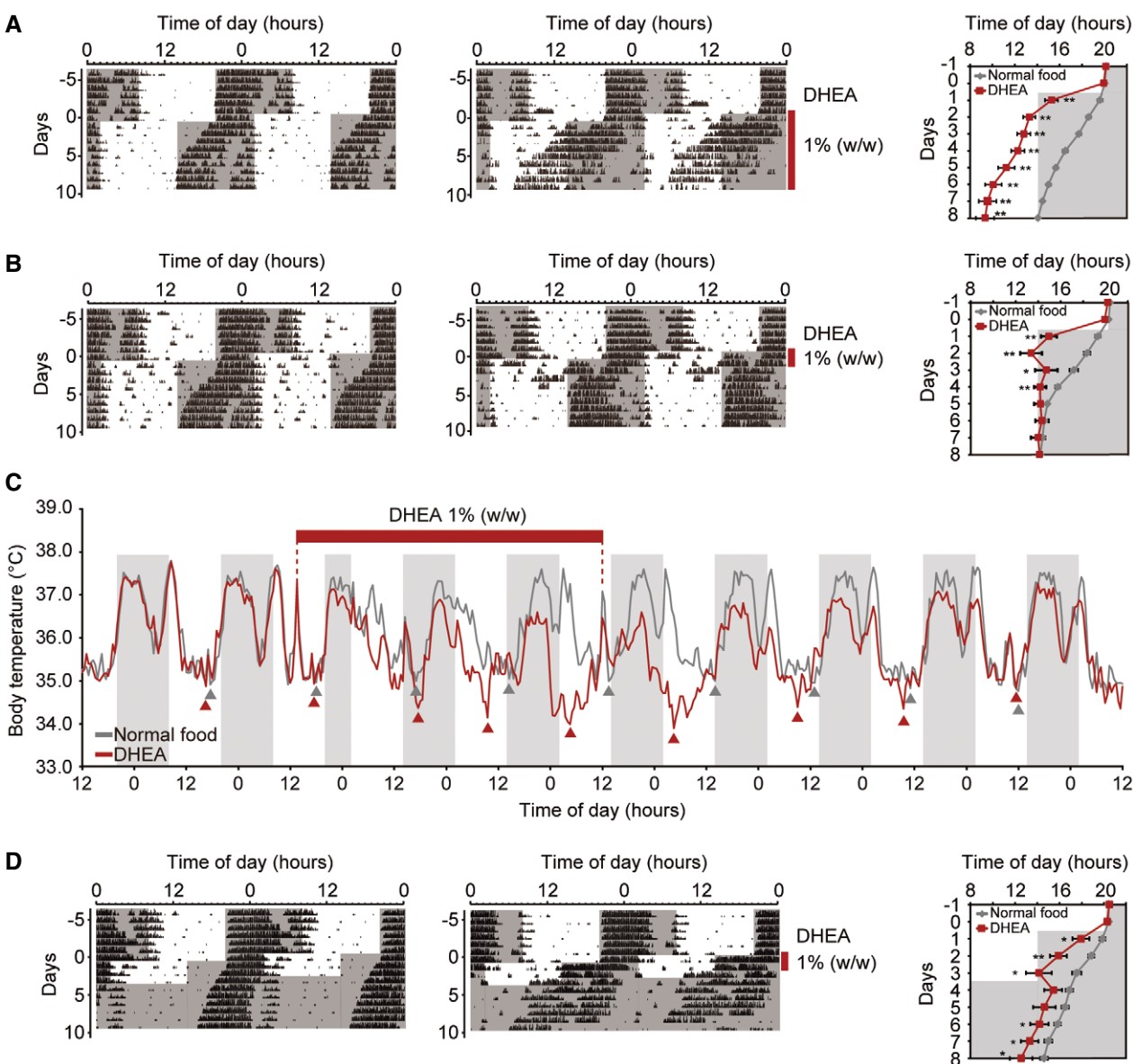

**Figure 3. Dietary administration of DHEA reduces jet-lag.**

A    Effect of chronic DHEA administration (1.0% w/w; vertical line) on wheel-running activity in mice during re-entrainment to a 6-h phase-advanced LD cycle. Representative double-plotted actograms of control ($n = 12$; left) and DHEA-treated ($n = 11$; middle) animals. Activity onset (mean ± SEM) during a 6-h phase advance was plotted (right panel) and analyzed by a Welch's $t$-test (**$P < 0.01$). All statistical information is shown in Appendix Table S1.

B    Effect of acute DHEA (3 days of 1.0% w/w) on wheel-running activity in mice during a 6-h phase-advanced LD cycle. Representative double-plotted actograms of control ($n = 10$; left) and DHEA-treated ($n = 11$; middle) animals. Activity onset during a 6-h phase-advanced LD cycle (mean ± SEM) (right panel). Data were analyzed by a Welch's $t$-test (*$P < 0.05$, **$P < 0.01$). All statistical information is shown in Appendix Table S1.

C    Effect of acute DHEA (3 days of 1.0% w/w; red bar above) on body temperature rhythms. After surgical implantation of temperature devices, body temperature was measured for the duration of the experiment. Data are shown for 2.5 days of entrainment to a LD cycle and then 7.5 days of exposure to a 6-h phase advance LD cycle. Data represent the mean of each treatment.

D    Effect of acute DHEA (3 days of 1.0% w/w) and exposure to a 6-h phase-advanced LD cycle (3 days) and then return to normal food and transfer into DD. Representative double-plotted actograms of control ($n = 10$; left) and DHEA-treated ($n = 11$; middle) animals. Activity onset was plotted during a 6-h phase-advanced LD cycle and then transfer into DD (mean ± SEM) (right panel). Data were analyzed by a Welch's $t$-test (*$P < 0.05$, **$P < 0.01$). All statistical information is shown in Appendix Table S1.

to jet-lag and immediately re-entrained to phase-shifted LD cycles. The authors proposed the crucial role of interneuronal communication within the SCN in this process. Indeed, they revealed by luminescent imaging of slice cultures that SCN neurons from mutant mice were more vulnerable to perturbation than those from wild-type mice, and it was this instability or lack of communication that enabled mutant mice to re-entrain more rapidly to new LD cycles. They also demonstrated that administration of V1a and V1b

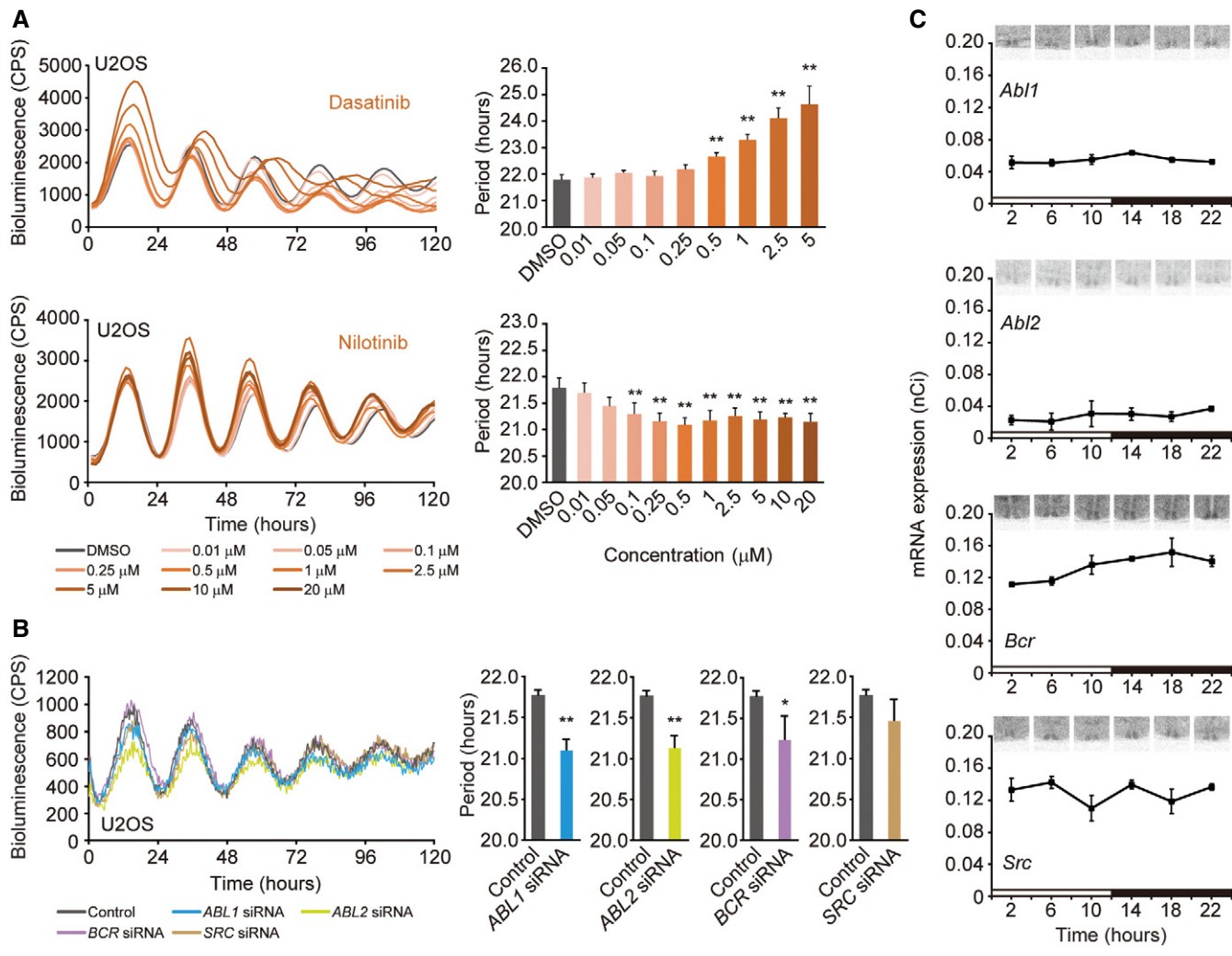

**Figure 4.  Inhibition of ABL and BCR kinases modulates circadian period.**

A   Dose-dependent effects of BCR-ABL tyrosine kinase inhibitors dasatinib and nilotinib on *Bmal1-dLuc* rhythms in U2OS cells. Luminescent traces from one of three or four independent experiments are shown. Circadian period was determined by curve fitting. Data are the mean ± SEM of three or four independent experiments and were analyzed by one-way ANOVA, followed by a Dunnett's test (**$P < 0.01$). All statistical information is shown in Appendix Table S1

B   Effect of siRNA-mediated knockdown of *ABL*, *BCR*, and *SRC* on circadian rhythms: *Bmal1-dLuc* rhythms (left) and circadian period (right). Data are presented as the mean ± SEM of three or four independent experiments (*$P < 0.05$, **$P < 0.01$, Welch's *t*-test). All statistical information is shown in Appendix Table S1.

C   Expression of *Abl1*, *Abl2*, *Bcr*, and *Src* in the mouse SCN by *in situ* hybridization. Data are presented as the mean ± SEM ($n = 3$) and were analyzed by least-squares regression with sinusoids. All statistical information is shown in Appendix Table S1.

antagonists directly into the SCN accelerated re-entrainment and proposed vasopressin receptors as therapeutic targets for the treatment of circadian misalignment due to travel or shift work (Yamaguchi *et al*, 2013). Similarly, administration of high concentrations of VIP into the SCN has been shown to reduce intercellular synchrony, increase phase-shifting, and accelerate re-entrainment in mice (An *et al*, 2013). This study also emphasized the role of decreased intercellular communication among SCN neurons in reducing jet-lag. Thus, SCN-specific chemicals and peptides that reduce intercellular communication appear to be effective in combating jet-lag (Yamaguchi *et al*, 2013; Doi *et al*, 2016). However, the development and administration of drugs that target the SCN specifically appears to be quite challenging.

It is well established that circadian clocks are present, not only in the SCN, but also in peripheral tissues and cells (Balsalobre *et al*, 1998; Yamazaki *et al*, 2000; Yoo *et al*, 2004). The SCN synchronizes and establishes an appropriate phase relationship with circadian clocks in the periphery. Therefore, to combat jet-lag, this phase relationship between the SCN and peripheral clocks must be maintained. However, this appears to be quite difficult because SCN and peripheral tissues have different sensitivities to different time cues (Wallach & Kramer, 2015). We hypothesize that period-changing drugs acting on multiple tissues may accelerate re-entrainment to phase-shifted LD cycles. Indeed, DHEA shortened circadian period of cells and explanted tissues, including the SCN and lung. Furthermore, dietary administration of DHEA shortened the circadian

period of wheel-running activity and body temperature rhythms under constant dark conditions. Although it is not yet clear whether the period-shortening and phase-advancing effects of DHEA were independent of the SCN, dietary administration dramatically accelerated re-entrainment to a 6-h phase-advanced LD cycle. These results suggest that period-changing drugs acting on peripheral tissues could aid in curing jet-lag. A chemical cocktail composed of both central and peripheral tissue-targeted drugs, such as DHEA, might therefore be a more effective treatment for jet-lag.

The mode of DHEA action is clearly of great interest. According to the DrugBank database (version 5.0.11 released 2017-12-20: www.drugbank.ca) (Wishart et al, 2018), there have been at least 12 targets of DHEA reported in humans (Table EV1). Among these, nuclear receptors, such as the androgen receptor (AR), the nuclear receptor subfamily 1 group I member 2 (NR1I2: pregnane X receptor, PXR) and 3 (NR1I3: constitutive androstane receptor, CAR), and the estrogen receptor β (ERβ), are of particular interest. Mice lacking Cry1 or Cry2 exhibit changes in free-running period (van der Horst et al, 1999), and recently, direct interactions between CRY1/2 proteins and these nuclear receptors have been reported (Kriebs et al, 2017). These data suggest a potential mechanism for DHEA action, as one of these nuclear receptors might be a direct target of DHEA and influence the regulation of circadian period by CRY1/2. It would therefore be interesting to test the effects of DHEA in mice lacking these nuclear receptors, as well as in mice lacking Cry1 or Cry2. An additional target for DHEA is the NMDA receptor (Table EV1), which is known to be involved in the phase-shifting effects of light on the circadian system (Colwell et al, 1991; Ding et al, 1994; Mintz et al, 1999; Asai et al, 2001). It is possible that DHEA might be affecting behavior by targeting the NMDA receptor and influencing this neural pathway. Interestingly, DHEA sulfate has been proposed to induce phase advances via $GABA_A$ receptors when administered during the subjective day (Pinto & Golombek, 1999), suggesting another possible mode of action for DHEA. Clearly, additional studies are required to identify the actual target (s) of DHEA responsible for reducing jet-lag in vivo.

Phosphorylation plays an important role in the circadian clock mechanism and regulates the stability and subcellular localization of clock proteins. Several protein kinases, including CK1δ, CK1ε, and CK2, AMP-activated kinase (AMPK), and glycogen synthase kinase 3β (GSK3β), have been shown to regulate circadian period and are therefore important for clock function (Hirota et al, 2008; Isojima et al, 2009; Chen et al, 2011; Takahashi, 2015). The vast majority of protein targets are phosphorylated on serine and threonine residues. Although tyrosine residues can also be phosphorylated, they constitute less than 1% of the total targets (Hunter, 2009). There are 90 tyrosine kinase genes in the human genome, but the physiological function of most of these proteins remains largely unknown. Using a chemical genetic approach, we discovered several tyrosine kinase inhibitors that affected circadian period and suggested the involvement of ABL and BCR protein kinases in the regulation of the mammalian circadian clock. The ABL1 gene is ubiquitously expressed, and Abl1-null mutant mice die within 1–2 weeks after birth (Schwartzberg et al, 1989; Tybulewicz et al, 1991). Therefore, it has not yet been possible to test the role of the ABL genes in circadian timekeeping using forward or reverse genetic approaches. To further understand the physiological function of the ABL1 and ABL2 genes, identification of their downstream targets is necessary, in addition to the generation of conditional knockouts of the ABL1 and/or ABL2 genes within the adult SCN.

By screening known bioactive compounds, we have discovered new circadian clock modulators. One compound, DHEA, shortens circadian period and reduces jet-lag in mice. Although further extensive testing is required, it is tempting to speculate that DHEA might be used for the treatment of jet-lag in humans. We have also suggested a role for non-receptor type tyrosine kinases ABL1, ABL2, and the BCR protein kinase in determining circadian period. Thus, drug repurposing is a useful approach to identify potential therapeutic treatments for circadian misalignment and understanding the underlying mechanisms of the circadian clock.

# Materials and Methods

### Chemicals

The SCREENWELL FDA-approved drug library V2 containing 741 compounds was purchased from Enzo Life Sciences (Hayashi Kasei Co., Ltd.), and the International Drug Collection (IDC) containing 311 compounds was purchased from MicroSource Discovery Systems, Inc. (Namiki Shoji Co., Ltd.). Aldosterone (A9477), DHEA (D4000), dehydroisoandrosterone 3-sulfate (D5297), and Reichstein's substance S (also called 11-deoxycortisol; R0500) were obtained from Sigma. Androstenedione (A0845), cholesterol (C0318), corticosterone (C0388), estriol (E0218), hydrocortisone (H0533), and stanolone (also known as dihydrotestosterone; A0462) were purchased from Tokyo Chemical Industry (TCI). 17-α hydroxypregnenolone (SC223186) and 21-hydroxyprogesterone (also called deoxycorticosterone; SC231274) were obtained from Santa Cruz Biotechnology. Tyrosine kinase inhibitors imatinib mesylate (sc-202180), nilotinib (sc-202245), dasatinib (sc-358114), bosutinib (sc-202084), bafetinib (sc-503249), and ponatinib (sc-362710) were purchased from Santa Cruz Biotechnology, Inc. Chemical stocks (10 mM in DMSO or ethanol for cholesterol) were either obtained from the manufacturer (chemical libraries) or prepared (individual chemicals) and stored at −20°C.

### Cell culture

Human U2OS cells containing a Bmal1-dLuc reporter gene were cultured as described (Oshima et al, 2015) in DMEM supplemented with 10% FBS, 2 mM L-glutamine, 100 unit/ml penicillin, and 100 μg/ml streptomycin. Mouse embryonic fibroblasts (MEFs) were prepared from heterozygous PER2::LUC mice at embryonic day 10 as described (Yagita et al, 2002). MEFs were cultured as described in DMEM supplemented with 10% FBS, 2 mM L-glutamine, 100 unit/ml penicillin, and 100 μg/ml streptomycin.

### Drug screening

U2OS cells were seeded at a density of 4,000 cells/well in 384-well plates and incubated for 2–3 days until confluent. Using a Caliper Life Sciences ALH 3000 Workstation, chemical library stocks were initially prepared at 5 and 0.5 mM in DMSO and then diluted to 10 and 1 μM, respectively, in assay media for measuring luminescence. This results in a final DMSO concentration of 0.2% in control and drug-treated samples. Luminescent assay media (also called "air" or

recording media) was composed of DMEM (Sigma D2902), 10 mM HEPES, 3.5 g/l D-glucose, 0.35 g/l sodium bicarbonate, 100 unit/ml penicillin, 100 μg/ml streptomycin, 2% B27 (Life Technologies), 0.1 mM luciferin (Wako), and 100 nM dexamethasone (Sigma). Diluted chemicals were pipetted into triplicate wells of a 384-well plate containing U2OS cells (prepared above), and bioluminescence was monitored for 5–6 days in a Churitsu CL384 Series luminometer (Churitsu Electric Corporation). Circadian period was determined using NINJA SL00-01 software for time series analysis (Churitsu Electric Corporation), and potential hit compounds were identified based on changes in circadian period. Drugs that lengthened circadian period by 1 or more hours or shortened period by 0.5 or more hours were selected. Compounds whose effects were clearly dose-dependent but below threshold values were also included as potential hits. These compounds were retested in a secondary screen for dose-dependency (0.5–20 μM in 0.2% DMSO).

Substrate availability has been shown to influence circadian parameters in luciferase reporter assays (Feeney *et al*, 2016). To ensure that the effects we observed with DHEA were robust and not due to insufficient substrate, we tested different concentrations of luciferin. Despite small increases in both luminescence and period with 0.2 mM compared to 0.1 mM luciferin, treatment with 20 μM DHEA consistently and significantly shortened circadian period (Appendix Fig S5A).

To verify that the effects of DHEA are independent of dexamethasone, corticosterone or other steroid hormones in the assay media, which might influence the circadian clock (Balsalobre *et al*, 2000a), we used charcoal-stripped fetal bovine serum, in which steroid hormones have been selectively removed (Thermo Fisher 12676029). In addition, we compared a 2-h dexamethasone pulse to tonic dexamethasone in the recording media. Despite a small increase in circadian period in cells pulsed with dexamethasone, our results show that 20 μM DHEA significantly shortened circadian period in U2OS cells under all conditions tested (Appendix Fig S5B). This indicates that the effects of DHEA are consistent and unlikely to be caused by perturbing the effects of steroids in the media.

To ensure that the effects of tyrosine kinase inhibitors dasatinib and nilotinib are not dependent on insulin in the media, which might affect the circadian clock (Balsalobre *et al*, 2000b), we prepared assay media with Xeno-free B27 supplement with or without insulin (Thermo Fisher A1486701, A3695201). At a concentration of 1 μM, dasatinib and nilotinib lengthened and shortened circadian period, respectively, in the presence or absence of insulin (Appendix Fig S5C). Thus, these two inhibitors appear to act on the circadian clock independently of insulin/insulin receptor tyrosine kinase.

### Explant cultures

Tissues were dissected from male and female *mPer2^Luc^* knockin mice, ~2–6 months old, and cultured essentially as described (Yoo *et al*, 2004) in 35-mm dishes containing Millicell cell culture inserts (Millipore) in assay media for measuring luminescence (same as above). Dishes were sealed with silicon grease and parafilm, and bioluminescence was measured in a LumiCycle 32 (Actimetrics) for 6 days. Circadian period was determined by using the LumiCycle Analysis software (Actimetrics), and samples with goodness of fit values over 80% were used in the analysis. *P*-values were determined by a Welch's *t*-test, and those below 0.05 were considered significant.

### The paper explained

**Problem**
Disruption of the circadian clock is becoming ever more common due to the prevalence of shift work and frequent travel across time zones. This leads to circadian desynchrony, or jet-lag, which reflects a mismatch between the internal biological clock and the environmental light–dark cycle. Chronic circadian misalignment has long-term consequences on our health and often leads to an increased risk of diabetes, cardiovascular disease and cancer. However, there are currently no orally available drugs, except for perhaps melatonin, to combat the complex behavioral and metabolic consequences of jet-lag.

**Results**
We used a high-throughput chemical screening strategy to isolate new circadian clock modulators, with the aim of repurposing or finding new functions for existing drugs. We discovered that the steroid hormone dehydroepiandrosterone (DHEA) significantly speeds up or shortens the period of the circadian clock in cells and tissues. When fed to mice, DHEA also shortened circadian period and significantly reduced jet-lag.

**Impact**
Although further studies are required, our results suggest that DHEA might serve as a convenient over-the-counter treatment for jet-lag.

### siRNA-mediated knockdown

Gene-specific packages of four preselected siRNAs to human *ABL1*, *ABL2*, *BCR*, *SRC*, and positive control *CRY2*, as well as a single negative control siRNA, were obtained from Qiagen (FlexiTube GeneSolution siRNA). These siRNAs were introduced singly, or as a mixture of two or four siRNAs into U2OS cells using a reverse transfection protocol in 96-well plates (Hirota *et al*, 2008) or 384-well plates (Zhang *et al*, 2009). Bioluminescence was monitored in an EnSpire Multimode Plate Reader (Perkin-Elmer) for 96-well plates or a Churitsu CL384 luminometer for 384-well plates. The TaqMan Gene Expression cells-to-Ct Kit (Ambion, Life Technologies) was used to lyse cells and reverse transcribe mRNA into cDNA. This cDNA was used for qPCR following the manufacturer's instructions (Applied Biosystems QuantStudio 3 Real-Time PCR system) using gene-specific TaqMan probes (Applied Biosystems Assay ID: Hs01104728_m1 for *ABL1*, Hs00943652_m1 for *ABL2*, Hs01036532_m1 for *BCR*, Hs1082246_m1 for *SRC*, Hs00323654_m1 for *CRY2*, Hs02758991_g1 for *GAPDH*). $\Delta C_t$ was determined using GAPDH as a reference gene, and relative expression was then calculated using the $\Delta\Delta C_t$ method by comparing gene-specific siRNA samples to the negative siRNA control.

### Behavioral analysis

Male *Mus musculus* C57BL/6J wild-type mice (8–10 weeks of age) were used in each experiment, with eight controls and 16 drug-treated mice, or 12 controls and 12 drug-treated mice. These mice were housed in individual cages equipped with running wheels, and activity was recorded using the Chronobiology Kit (Stanford Software Systems, Stanford, CA). Cages were placed in light-tight boxes, where light was provided by fluorescent lamps (Panasonic FHF32EX-N-H; 1,000–2,000 lux at the top of the cage). Mice were maintained on a 12:12 LD cycle, unless otherwise indicated, and fed

~50 g of powdered food (with or without drug) weekly. DHEA powder was added directly to powdered food at a concentration of 0.5% (w/w) or 1.0% (w/w) and administered at noon ~1 week after transfer into DD, or the day before a 6-h phase advance, where the dark phase was shortened by 6 h. Body temperature was measured using temperature data loggers (Thermochron type-SL; KN Laboratories) surgically implanted into mice, and the data were retrieved using RhManager software (KN Laboratories). This study was approved by the Animal Experiment Committee of Nagoya University.

### *In situ* hybridization

Mice were maintained for several days on a 12:12 LD cycle. At the indicated zeitgeber time (ZT), mice were killed and brains were dissected and frozen immediately on dry ice. Non-perfused frozen sections (20 μm) were prepared and probed with $^{33}$P-labeled oligonucleotides (Yoshimura *et al*, 2000). Three to four oligonucleotides were designed for each gene and used to increase sensitivity. Hybridization was carried out overnight at 42°C, and two high-stringency post-hybridization washes were performed at 55°C. Sections were air-dried and exposed to BioMax MR film (Eastman Kodak). Listed below are the probe sequences for each gene examined:

*Abl1*  tccttggagttccatcgagctgcttcgctgagaccctggggctca;
catgtagagcagtaccacggcgctcacctcctgccggttacactc;
gccggtcagagggggttccactgccaacatgctcgcatgagctcgt;
tctcagtccttctccaggtgggccggccttggtggggtcagtgtt

*Abl2*  tgatcgtgctgggtgaagacgttgaagccggcgtctgcggtacgc;
ctcagggtcctcgtcttggaaggaagcagcggtaaccgaggcagg;
cggagagtcactctctgtgatggctggatcaggggctctcaaggg

*Bcr*  tcgctggaggtgaggttctcgttggagctgcagtccggcgtgtag;
cagctctaccataaagctgtccttcagcagctgccggtgctctcc;
ctgctgctctcggatgctttctctccactcggctcgctcatagtc;
actcggtagatgcccacctcctccatacccggcgctcgatctct

*Src*  gaggtgacggtgtccgaggagttgaagcctccgaagagcttgggc;
aaggtccctctcgggttctcggcgttgagcagcagccgctctgat;
gcctcctgcaggaaggcctctggggacatggtgcctggcttcaga;
ggaccacacatccgacttgatggtgaacctgccgtacagagcagc

### Statistical analysis

Data are presented as the mean ± standard error of the mean (SEM). Data were analyzed according to the statistical tests specified in the figure legends and Appendix Tables S1 and S2. Exact *P*-values for all comparisons are available in Appendix Tables S1 and S2.

**Expanded View** for this article is available online.

### Acknowledgements

We thank Dr. Joseph S. Takahashi for providing *mPer2$^{Luc}$* knockin mice and Drs. Daisuke Ono and Tsuyoshi Hirota for technical advice. This work was supported by the JSPS KAKENHI "Grant-in-Aid for Specially Promoted Research" (Grant Number 26000013) and "Grant-in-Aid for Young Scientists (B)" (Grant 17K17797), Human Frontier Science Program (RGP0030/2015), and Research Foundation for Oriental Medicine. WPI-ITbM is supported by the World Premier International Research Center Initiative (WPI), MEXT, Japan.

## Author contributions

TY conceived and designed research; TKT, YN, WO, AK, MI, TNO, and TY performed research and analyzed data; NK, KI, KY, YS, MS, and AS contributed new reagents; TKT and TY wrote the article; all authors commented on the article.

## Conflict of interest

The authors declare that they have no conflict of interest.

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
