## [Review Process File · EMBO Molecular Medicine]

Identification of circadian clock modulators from existing drugs

T. Katherine Tamai, Yusuke Nakane, Wataru Ota, Akane Kobayashi, Masateru Ishiguro, Naoya Kadofusa, Keisuke Ikegami, Kazuhiro Yagita, Yasufumi Shigeyoshi, Masaki Sudo, Taeko Nishiwaki-Ohkawa, Ayato Sato & Takashi Yoshimura

Review timeline:	Submission date:	26 November 2017
	Editorial Decision:	10 January 2018
	Revision received:	19 February 2018
	Editorial Decision:	6 March 2018
	Revision received:	19 March 2018
	Accepted:	22 March 2018

Editor: Céline Carret

Transaction Report:

1st Editorial Decision

10 January 2018

Thank you for the submission of your manuscript to EMBO Molecular Medicine. We have now heard back from the two referees whom we asked to evaluate your manuscript.

You will see from the set of reports pasted below that both referees are enthusiastic about the study and only have minor comments: both agree that a discussion on the *in vivo* mechanism is needed [and I would like to encourage you to provide additional experimental evidence if you already have some data]. Referee 1 suggests a couple of controls to be performed and an *in vivo* experiment to strengthen the findings. Both referees have additional points that should be addressed as suggested.

We would welcome the submission of a revised version within three months for further consideration and would like to encourage you to address all the criticisms raised as suggested to improve conclusiveness and clarity. Depending on the nature of the revision, the manuscript may be sent back to the referees. Please note that EMBO Molecular Medicine strongly supports a single round of revision and that, as acceptance or rejection of the manuscript will depend on another round of review, your responses should be as complete as possible.

I look forward to receiving your revised manuscript.

***** Reviewer's comments *****

Referee #1 (Remarks for Author):

This is a solid observational study, which will provide a useful resource for the circadian research field and has some clear translational potential in the longterm. The authors are to be commended for their clear communication of intriguing experimental findings, it disappointed me that little attention is given to the mode of action by which many of the drugs that affect circadian rhythms in their assays might elicit their effects. Overall I would certainly recommend that this manuscript be accepted for publication, however there are several technical issues that first should really be

addressed:

1) Previous studies - with regards to DHEA, more effort must be made to interpret these data in light of prior observations by other labs using DHEA and its effect on the amplitude and period of circadian gene expression rhythms. In particular, using 50 μ M DHEA also with human U2OS cells, Rey et al (Cell metabolism, 2016) observed reversible lengthening of circadian rhythms reported by Bmal1:Luc as well as a reduction of reporter amplitude. Whereas Putker et al (ARS, 2017) report that DHEA both lengthened and shortened the period of PER2::LUC rhythms in primary mouse fibroblasts, dependent on concentration, but found that it consistently reduced the amplitude of clock gene bioluminescent reporter rhythms across several different cell types.

2) Effects on circadian reporter vs. effects on circadian rhythms - all the in vitro studies rely on firefly luciferase as a reporter of circadian gene expression. Unless the concentration of luciferin intracellularly is in saturating excess, it is plausible that subtle changes in apparent circadian period (e.g. Fig. 4B, Fig2A) actually result from drug-induced changes in intracellular luciferin availability (Patrick et al., Mol Imaging Biol, 2014; Feeney et al., JBR, 2016). To rule out this possibility, an acute increase in extracellular luciferin concentration (e.g. from 0.1 mM to 0.2 mM) would not be expected to produce any increase in bioluminescence, and at the very least if it does it should not be affected by any of the drugs in question. This simple control experiment should be performed for each cell/explant model that is employed and be reported in supplementary data, to reassure the reader that the apparent effects of DHEA and other drugs on circadian rhythms in cells and tissue explants is not specific to these experimental conditions and reporter system.

3) Drug interactions with media components - The authors use 100 nM dexamethasone in their recording media as well as the B27 supplement, which when diluted to 2% I believe will give final concentrations of 58 nM corticosterone and 600 nM insulin (Chen et al., J Neurosci Methods, 2009). Dexamethasone, corticosterone and insulin themselves have the capacity to affect circadian rhythms in cultured cells and tissues, acting via steroid hormone receptors and insulin receptors respectively (e.g. Basalobre et al., Curr Biol and Science, 2000). At the relatively high concentrations used in this study, as a steroid hormone in principal DHEA may elicit its effects by perturbing the effect of dexamethasone and/or corticosterone rather having any direct effect on cellular rhythms itself. Similarly, tyrosine kinase inhibitors may appear to affect cellular rhythms by antagonizing the affect that high extracellular insulin would otherwise exert. To discount this possibility it would be prudent to assess the effect of DHEA, dasatinib and nicotinib upon circadian rhythms in U2OS cells recorded using media that lacks corticosterone, dexamethasone and insulin. Again this is a quick simple control that should be reported as a supplementary figure which will lend confidence to the reader that the action of drugs identified by this screen is not specific to the particular culture conditions employed for bioluminescence recording assays.

4) Solvent effects - the final concentration of DMSO used in the cellular assays should be reported in the methods section, since at sufficiently high concentrations (>0.5%) it will affect circadian rhythms as well as luciferase activity.

5) In vivo experiments - the in vivo experiments using DHEA are really quite striking, however in the actogram presented in Figure 2C it appears to me that during DHEA treatment, whilst the onset of activity clearly advances compared with controls the same may not be true of the activity offset. As such, it is plausible that DHEA is changing the organization of behavior under constant darkness rather than truly shortening the period of the endogenous clock. Either possibility would be most interesting, but to distinguish between them it would be sensible to quantify how the timing of both activity offset and onset in the DHEA group compares with controls.

6) DHEA targets - a brief survey of the extant literature suggests a bewildering array of potential mechanisms through which DHEA may exert its effects in rodents in vivo. Even the Wikipedia page for DHEA lists multiple references for the following: androgen receptors, estrogen receptors, GABAA receptors, NMDA receptors, PPARalpha and PXR nuclear receptors, as well as inhibition of CYP2C11 and 11Beta-HSD1 which are essential for glucocorticoid biosynthesis. My money would be on the latter being the basis of DHEA's effects in vivo, since it is easy to conceive how suppression of SCN/HPA-dependent corticosterone-mediated timing cues would render mice more sensitive to feeding cues allowing all cellular clocks outside the SCN to reset more rapidly (e.g. Saini et al., Genes Dev., 2013). I would not expect the authors to test this of course, but I think the

reader would appreciate it if the discussion section made some explicit and testable predictions about how the effects of DHEA on circadian rhythms in vivo might be mechanistically delineated.

Referee #2 (Remarks for Author):

This article by Tamai and colleagues describes the identification of drugs able to modulate the phase of a circadian clock reporter in U2OS cells. Among the identified drugs, the steroid hormone DHEA, and other derived steroid hormones, presents a dose-dependent robust phase-shortening effect that is conserved in mouse. As a consequence, acute treatment with DHEA reduces jet-lag effect in mouse through fastening of the resynchronization, opening the possibility to use the drug in human. Other drugs affecting the BCR and ABL kinases, as well as knockdown of the respective genes, also affect the period length of U2OS cells, validating the approach of the discovery of new pathways modulating the phase of the circadian clock.

This is a very well conducted interesting study that give a new example of potential treatment for fast resynchronization after jet-lag. However, as discussed by authors, application to human is still very hypothetical. The weak point of the article is the absence of the description of the involved molecular mechanisms for the 2 selected examples. Although such question is out of the scope of the study, some discussion about the potential DHEA receptor involved or the link between the BCR-ABL kinases and the cell cycle described also to affect period length (for example PMID 25028488 or 24958884) will increase the quality of the discussion.

1st Revision - authors' response

19 February 2018

Referee #1

This is a solid observational study, which will provide a useful resource for the circadian research field and has some clear translational potential in the long term. The authors are to be commended for their clear communication of intriguing experimental findings, it disappointed me that little attention is given to the mode of action by which many of the drugs that affect circadian rhythms in their assays might elicit their effects. Overall I would certainly recommend that this manuscript be accepted for publication, however there are several technical issues that first should really be addressed:

Response: To indicate the possible mode of action of our hit compounds, we have listed all known targets in Table EV1 and added a histogram of the top six categories of targets (Fig 1C). We have also added the following text to the Results section: "One obvious advantage of screening existing drugs is that they frequently have known mechanisms of action (Rennekamp & Peterson, 2015). All known targets of hit compounds identified in our screen are listed in Table EV1. In addition, the top six categories of drug targets are shown and include DNA, the androgen receptor (AR), DNA topoisomerase 2 α , the nuclear receptor subfamily 1 group I member 2 (NR1I2), the retinoic acid receptor α (RAR α)/retinoid X receptor β (RXR β), and the tubulin β chain (Fig 1C)."

1) Previous studies - with regards to DHEA, more effort must be made to interpret these data in light of prior observations by other labs using DHEA and its effect on the amplitude and period of circadian gene expression rhythms. In particular, using 50 μ M DHEA also with human U2OS cells, Rey et al (Cell metabolism, 2016) observed reversible lengthening of circadian rhythms reported by Bmal1:Luc as well as a reduction of reporter amplitude. Whereas Putker et al (ARS, 2017) report that DHEA both lengthened and shortened the period of PER2::LUC rhythms in primary mouse fibroblasts, dependent on concentration, but found that it consistently reduced the amplitude of clock gene bioluminescent reporter rhythms across several different cell types.

Response: Previous circadian drug screens have identified compounds that, depending on dose, have opposite effects on circadian period (Hirota et al., 2008). As pointed out by referee 1, this might also be the case for DHEA. To confirm this, we examined the effect of higher concentrations of DHEA on circadian period in U2OS cells. Treatment with 100 μ M DHEA

clearly and significantly lengthened period and reduced amplitude compared to DMSO controls (Fig EV1), as previously reported (Rey et al., 2016; Putker et al., 2017). Although 50 μ M DHEA also reduced amplitude, this treatment actually shortened circadian period in U2OS cells in our hands (Fig EV1). This was also true for MEFs and explants of SCN and lung, as shown in Figure 2. Perhaps 50 μ M is approaching the critical concentration required for the period-switching effect of DHEA, but is not quite high enough under our assay conditions. We have added this new information in Figure EV1 and have added the following text: “Recent reports, however, have shown that high doses of DHEA (e.g., 100 μ M) lengthen circadian period and decrease amplitude (Rey et al, 2016; Putker et al, 2017), which we have confirmed (Fig EV1).”

2) Effects on circadian reporter vs. effects on circadian rhythms - all the in vitro studies rely on firefly luciferase as a reporter of circadian gene expression. Unless the concentration of luciferin intracellularly is in saturating excess, it is plausible that subtle changes in apparent circadian period (e.g. Fig. 4B, Fig2A) actually result from drug-induced changes in intracellular luciferin availability (Patrick et al., Mol Imaging Biol, 2014; Feeney et al., JBR, 2016). To rule out this possibility, an acute increase in extracellular luciferin concentration (e.g. from 0.1 mM to 0.2 mM) would not be expected to produce any increase in bioluminescence, and at the very least if it does it should not be affected by any of the drugs in question. This simple control experiment should be performed for each cell/explant model that is employed and be reported in supplementary data, to reassure the reader that the apparent effects of DHEA and other drugs on circadian rhythms in cells and tissue explants is not specific to these experimental conditions and reporter system.

Response: To ensure that the effects we observed with DHEA were robust and not due to insufficient substrate in our luciferase assay (drug-induced or otherwise), we tested different concentrations of luciferin (0.1 mM and 0.2 mM) in the media. Despite small increases in luminescence and circadian period with 0.2 mM compared to 0.1 mM luciferin, treatment with 20 μ M DHEA consistently and significantly shortened circadian period. These results are shown in Appendix Figure S5A, and the following text has been added to the Methods section: “Substrate availability has been shown to influence circadian parameters in luciferase reporter assays (Feeney et al, 2016). To ensure that the effects we observed with DHEA were robust and not due to insufficient substrate, we tested different concentrations of luciferin. Despite small increases in both luminescence and period with 0.2 mM compared to 0.1 mM luciferin, treatment with 20 μ M DHEA, consistently and significantly shortened circadian period (Appendix Fig S5A).”

3) Drug interactions with media components - The authors use 100 nM dexamethasone in their recording media as well as the B27 supplement, which when diluted to 2% I believe will give final concentrations of 58 nM corticosterone and 600 nM insulin (Chen et al., J Neurosci Methods, 2009). Dexamethasone, corticosterone and insulin themselves have the capacity to affect circadian rhythms in cultured cells and tissues, acting via steroid hormone receptors and insulin receptors respectively (e.g. Basalobre et al., Curr Biol and Science, 2000). At the relatively high concentrations used in this study, as a steroid hormone in principal DHEA may elicit its effects by perturbing the effect of dexamethasone and/or corticosterone rather having any direct effect on cellular rhythms itself. Similarly, tyrosine kinase inhibitors may appear to affect cellular rhythms by antagonizing the affect that high extracellular insulin would otherwise exert. To discount this possibility it would be prudent to assess the effect of DHEA, dasatinib and nicotinib upon circadian rhythms in U2OS cells recorded using media that lacks corticosterone, dexamethasone and insulin. Again this is a quick simple control that should be reported as a supplementary figure which will lend confidence to the reader that the action of drugs identified by this screen is not specific to the particular culture conditions employed for bioluminescence recording assays.

Response: To test whether DHEA shortens circadian period by perturbing the effects of dexamethasone, corticosterone or other steroid hormones in our assay media, we used charcoal-stripped fetal bovine serum, in which steroid hormones have been selectively removed. Since we typically use dexamethasone to synchronize the circadian clocks in our cells, we also compared a two-hour dexamethasone pulse to tonic dexamethasone in the recording media (our usual protocol). Despite a small increase in circadian period in cells pulsed with dexamethasone, the results show that 20 μ M DHEA significantly shortened circadian period in U2OS cells under all conditions tested (Appendix Fig S5B). This indicates

that the effects of DHEA are consistent and unlikely to be caused by perturbing the effects of steroids in the media. Notably, when we examined the effect of several steroid hormones on circadian period in U2OS cells (Appendix Fig S2), we showed that 1 μ M and 10 μ M corticosterone had no effect on circadian period. In fact, DHEA was the only endogenous steroid hormone tested that significantly shortened circadian period (Appendix Fig S2).

To examine whether Dasatinib or Nilotinib exert their effects on the circadian clock by antagonizing the effects of insulin in the media, we used Xeno-free B27 supplement with or without insulin (ThermoFisher A1486701, A3695201). At a concentration of 1 μ M, Dasatinib and Nilotinib lengthened and shortened circadian period, respectively, in the presence or absence of insulin (Appendix Fig S5C). This is consistent with our results in Figure 4 and shows that these two tyrosine-kinase inhibitors appear to act on the circadian clock independently of insulin/insulin receptor tyrosine kinase. Furthermore, the results from our siRNA experiments support the hypothesis that these drugs are acting through the ABL1/2 and BCR targets (see Fig 4B).

We have summarized this information by adding the following text to the Methods section: "To verify that the effects of DHEA are independent of dexamethasone, corticosterone or other steroid hormones in the assay media, which might influence the circadian clock (Balsalobre et al, 2000a), we used charcoal-stripped fetal bovine serum, in which steroid hormones have been selectively removed (ThermoFisher 12676029). In addition, we compared a two-hour dexamethasone pulse to tonic dexamethasone in the recording media. Despite a small increase in circadian period in cells pulsed with dexamethasone, our results show that 20 μ M DHEA significantly shortened circadian period in U2OS cells under all conditions tested (Appendix Fig S5B). This indicates that the effects of DHEA are consistent and unlikely to be caused by perturbing the effects of steroids in the media.

To ensure that the effects of tyrosine kinase inhibitors Dasatinib and Nilotinib are not dependent on insulin in the media, which might affect the circadian clock (Balsalobre et al, 2000b), we prepared assay media with Xeno-free B27 supplement with or without insulin (ThermoFisher A1486701, A3695201). At a concentration of 1 μ M, Dasatinib and Nilotinib lengthened and shortened circadian period, respectively, in the presence or absence of insulin (Appendix Fig S5C). Thus, these two inhibitors appear to act on the circadian clock independently of insulin/insulin receptor tyrosine kinase."

4) Solvent effects - the final concentration of DMSO used in the cellular assays should be reported in the methods section, since at sufficiently high concentrations (>0.5%) it will affect circadian rhythms as well as luciferase activity.

Response: In our drug screening and luminescent assays, we routinely use 0.2 % DMSO as a control for drug-treated cells. We have included this information in the Methods section, as shown below, and changed the text to clarify this point.

"U2OS cells were seeded at a density of 4000 cells/well in 384-well plates and incubated for 2-3 days until confluent. Using a Caliper Life Sciences ALH 3000 Workstation, chemical library stocks were initially prepared at 5 mM and 0.5 mM in DMSO and then diluted to 10 μ M and 1 μ M, respectively, in assay media for measuring luminescence. This resulted in a final concentration of 0.2% DMSO in control and drug-treated samples. Luminescent assay media (also called "air" or recording media) was composed of DMEM (Sigma D2902), 10 mM HEPES, 3.5 g/L D-glucose, 0.35 g/L sodium bicarbonate, 100 unit/mL penicillin, 100 μ g/mL streptomycin, 2% B27 (Life Technologies), 0.1 mM luciferin (Wako) and 100 nM dexamethasone (Sigma). Diluted chemicals were pipetted into triplicate wells of a 384-well plate containing U2OS cells (prepared above), and bioluminescence was monitored for 5-6 days in a Churitsu CL384 Series luminometer (Churitsu Electric Corporation). Circadian period was determined using NINJA SL00-01 software for time series analysis (Churitsu Electric Corporation), and potential hit compounds were identified based on changes in circadian period. Drugs that lengthened circadian period by one or more hours or shortened period by 0.5 or more hours were selected. Compounds whose effects were clearly dose-dependent, but below threshold values were also included as potential hits. These compounds were retested in a secondary screen for dose-dependency (0.5 μ M to 20 μ M in 0.2% DMSO)."

5) In vivo experiments - the in vivo experiments using DHEA are really quite striking, however in the actogram presented in Figure 2C it appears to me that during DHEA treatment, whilst the onset of activity clearly advances compared with controls the same may not be true of the activity offset.

As such, it is plausible that DHEA is changing the organization of behavior under constant darkness rather than truly shortening the period of the endogenous clock. Either possibility would be most interesting, but to distinguish between them it would be sensible to quantify how the timing of both activity offset and onset in the DHEA group compares with controls.

Response: As suggested, we have calculated circadian period based on activity offset and find that it, too, is significantly shortened in DHEA-treated animals compared to controls. These results have been added as panel C to Fig 2.

6) DHEA targets - a brief survey of the extant literature suggests a bewildering array of potential mechanisms through which DHEA may exert its effects in rodents *in vivo*. Even the Wikipedia page for DHEA lists multiple references for the following: androgen receptors, estrogen receptors, GABAA receptors, NMDA receptors, PPARalpha and PXR nuclear receptors, as well as inhibition of CYP2C11 and 11Beta-HSD1 which are essential for glucocorticoid biosynthesis. My money would be on the latter being the basis of DHEA's effects *in vivo*, since it is easy to conceive how suppression of SCN/HPA-dependent corticosterone-mediated timing cues would render mice more sensitive to feeding cues allowing all cellular clocks outside the SCN to reset more rapidly (e.g. Saini et al., *Genes Dev.*, 2013). I would not expect the authors to test this of course, but I think the reader would appreciate it if the discussion section made some explicit and testable predictions about how the effects of DHEA on circadian rhythms *in vivo* might be mechanistically delineated.

Response: To address the potential mechanism of DHEA action, we have added the following text to the Discussion section: “The mode of DHEA action is clearly of great interest. According to the DrugBank database (version 5.0.11 released 2017-12-20: www.drugbank.ca) (Wishart et al, 2017), there have been at least 12 targets of DHEA reported in humans (Table EV1). Among these, nuclear receptors, such as the androgen receptor (AR), the nuclear receptor subfamily 1 group I member 2 (NR1I2: pregnane X receptor, PXR) and 3 (NR1I3: constitutive androstane receptor, CAR), and the estrogen receptor β (ER β), are of particular interest. Mice lacking *Cry1* or *Cry2* exhibit changes in free-running period (van der Horst et al, 1999), and recently, direct interactions between CRY1/2 proteins and these nuclear receptors has been reported (Kriebs et al, 2017). These data suggest a potential mechanism for DHEA action, as one of these nuclear receptors might be a direct target of DHEA and influence the regulation of circadian period by CRY1/2. It would therefore be interesting to test the effects of DHEA in mice lacking these nuclear receptors, as well as in mice lacking *Cry1* or *Cry2*. An additional target for DHEA is the NMDA receptor (Table EV1), which is known to be involved in the phase-shifting effects of light on the circadian system (Colwell et al, 1991; Ding et al, 1994; Mintz et al, 1999; Asai et al, 2001). It is possible that DHEA might be affecting behavior by targeting the NMDA receptor and influencing this neural pathway. Interestingly, DHEA sulfate has been proposed to induce phase-advances via GABA_A receptors when administered during the subjective day (Pinto & Golombek, 1999), suggesting another possible mode of action for DHEA. Clearly, additional studies are required to identify the actual target(s) of DHEA responsible for reducing jet-lag *in vivo*.”

Referee #2:

This article by Tamai and colleagues describes the identification of drugs able to modulate the phase of a circadian clock reporter in U2OS cells. Among the identified drugs, the steroid hormone DHEA, and other derived steroid hormones, presents a dose-dependent robust phase-shortening effect that is conserved in mouse. As a consequence, acute treatment with DHEA reduces jet-lag effect in mouse through fastening of the resynchronisation, opening the possibility to use the drug in human. Other drugs affecting the BCR and ABL kinases, as well as knockdown of the respective genes, also affect the period length of U2OS cells, validating the approach of the discovery of new pathways modulating the phase of the circadian clock.

This is a very well conducted interesting study that give a new example of potential treatment for fast resynchronization after jet-lag. However, as discussed by authors, application to human is still very hypothetical. The weak point of the article is the absence of the description of the involved molecular mechanisms for the 2 selected examples. Although such question is out of the scope of the study, some discussion about the potential DHEA receptor involved or the link between the BCR-ABL kinases and the cell cycle described also to affect period length (for example PMID 25028488 or 24958884) will increase the quality of the discussion.

Response: Although a potential link between BCR-ABL kinases and the cell cycle in terms of regulating circadian period is interesting, we think it might be a bit premature to mention this in the discussion. However, we have added the following text addressing the possible mode of DHEA action in the Discussion section as suggested: “The mode of DHEA action is clearly of great interest. According to the DrugBank database (version 5.0.11 released 2017-12-20: www.drugbank.ca) (Wishart et al, 2017), there have been at least 12 targets of DHEA reported in humans (Table EV1). Among these, nuclear receptors, such as the androgen receptor (AR), the nuclear receptor subfamily 1 group I member 2 (NR1I2: pregnane X receptor, PXR) and 3 (NR1I3: constitutive androstane receptor, CAR), and the estrogen receptor β (ER β), are of particular interest. Mice lacking *Cry1* or *Cry2* exhibit changes in free-running period (van der Horst et al, 1999), and recently, direct interactions between CRY1/2 proteins and these nuclear receptors has been reported (Kriebs et al, 2017). These data suggest a potential mechanism for DHEA action, as one of these nuclear receptors might be a direct target of DHEA and influence the regulation of circadian period by CRY1/2. It would therefore be interesting to test the effects of DHEA in mice lacking these nuclear receptors, as well as in mice lacking *Cry1* or *Cry2*. An additional target for DHEA is the NMDA receptor (Table EV1), which is known to be involved in the phase-shifting effects of light on the circadian system (Colwell et al, 1991; Ding et al, 1994; Mintz et al, 1999; Asai et al, 2001). It is possible that DHEA might be affecting behavior by targeting the NMDA receptor and influencing this neural pathway. Interestingly, DHEA sulfate has been proposed to induce phase-advances via GABA_A receptors when administered during the subjective day (Pinto & Golombek, 1999), suggesting another possible mode of action for DHEA. Clearly, additional studies are required to identify the actual target(s) of DHEA responsible for reducing jet-lag *in vivo*.”

2nd Editorial Decision

6 March 2018

Thank you for the submission of your revised manuscript to EMBO Molecular Medicine. We have now received the enclosed reports from the referees that were asked to re-assess it. As you will see the reviewers are now globally supportive and I am pleased to inform you that we will be able to accept your manuscript pending the following final amendments:

1) Please address the comments made by referee 2. Please provide a letter INCLUDING my comments and the reviewer's reports and your detailed responses to their comments (as Word file).

***** Reviewer's comments *****

Referee #1 (Remarks for Author):

I have little to add to my previous positive remarks and am most impressed by the revised manuscript, which completely addresses all the issues I perceived in the original submission. I strongly recommend publication.

Referee #2 (Remarks for Author):

While the main comment by reviewers was the lack of description of the biochemical mechanisms involved in the action of the drugs, authors gave only few additional information regarding that aspect. Nevertheless, I recommend the acceptance of this interesting article.

However, I have a problem with this paragraph added by authors in the revised version:

"Among these, nuclear receptors, such as the androgen receptor (AR), the nuclear receptor subfamily 1 group I member 2 (NR1I2: pregnane X receptor, PXR) and 3 (NR1I3: constitutive androstane receptor, CAR), and the estrogen receptor β (ER β), are of particular interest. Mice lacking *Cry1* or *Cry2* exhibit changes in free-running period (van der Horst et al, 1999), and recently, direct interactions between CRY1/2 proteins and these nuclear receptors has been reported (Kriebs et al, 2017). These data suggest a potential mechanism for DHEA action, as one of these nuclear receptors might be a direct target of DHEA and influence the regulation of circadian period by

CRY1/2. It would therefore be interesting to test the effects of DHEA in mice lacking these nuclear receptors, as well as in mice lacking Cry1 or Cry2."

While this article by Kriebset and colleagues indeed described the interaction and regulation of the aforementioned nuclear receptors and CRY proteins, there is no evidence on the regulation of Cry genes and circadian period by these nuclear receptors. This paragraph is therefore purely speculative and not based on any data. Consequently, I am suggesting to remove it.

2nd Revision - authors' response

19 March 2018

While we are grateful for another opportunity to improve our manuscript, we are concerned that the discussion we added to address one referee has put us slightly at odds with the other. We acknowledge that this section is speculative, but we would prefer to keep the text in question for the reasons stated below.

Referee #1 (Remarks for Author):

I have little to add to my previous positive remarks and am most impressed by the revised manuscript, which completely addresses all the issues I perceived in the original submission. I strongly recommend publication.

Response: We are happy that referee 1 is satisfied with the revised version of our manuscript. Referee 2, however, has suggested removing a section of our discussion in which we have addressed comments by referee 1. If possible, we would like to keep this section on the interaction between nuclear receptors and cryptochromes, and our speculation that this might influence circadian period, since cryptochromes are well known to be involved in regulating period.

Referee #2 (Remarks for Author):

While the main comment by reviewers was the lack of description of the biochemical mechanisms involved in the action of the drugs, authors gave only few additional information regarding that aspect. Nevertheless, I recommend the acceptance of this interesting article. However, I have a problem with this paragraph added by authors in the revised version:

"Among these, nuclear receptors, such as the androgen receptor (AR), the nuclear receptor subfamily 1 group 1 member 2 (NR1I2: pregnane X receptor, PXR) and 3 (NR1I3: constitutive androstane receptor, CAR), and the estrogen receptor β (ER β), are of particular interest. Mice lacking Cry1 or Cry2 exhibit changes in free-running period (van der Horst et al, 1999), and recently, direct interactions between CRY1/2 proteins and these nuclear receptors has been reported (Kriebset al, 2017). These data suggest a potential mechanism for DHEA action, as one of these nuclear receptors might be a direct target of DHEA and influence the regulation of circadian period by CRY1/2. It would therefore be interesting to test the effects of DHEA in mice lacking these nuclear receptors, as well as in mice lacking Cry1 or Cry2."

While this article by Kriebset and colleagues indeed described the interaction and regulation of the aforementioned nuclear receptors and CRY proteins, there is no evidence on the regulation of Cry genes and circadian period by these nuclear receptors. This paragraph is therefore purely speculative and not based on any data. Consequently, I am suggesting to remove it.

Response: We unfortunately do not know the biochemical mechanism by which DHEA is acting on the circadian clock, but it is for this reason that we can only speculate, and we fully acknowledge this. Referee 2 is correct in stating that there is currently no evidence that the interaction of nuclear receptors with Cry proteins influences circadian period, but it is well known that Cry proteins themselves are involved in regulating period. As referee 1 requested explicit and testable predictions for DHEA action, we proposed examining the effects of DHEA in mice lacking specific nuclear receptors and/or mice lacking Cry1 or Cry2, which seems reasonable. We would therefore prefer to keep the added text, if possible.

YOU MUST COMPLETE ALL CELLS WITH A PINK BACKGROUND ↓
PLEASE NOTE THAT THIS CHECKLIST WILL BE PUBLISHED ALONGSIDE YOUR PAPER

Corresponding Author Name: Takashi Yoshimura
Journal Submitted to: EMBO Molecular Medicine
Manuscript Number: EMM-2017-08724